# Management of Renal Angiomyolipomas in Tuberous Sclerosis Complex: A Case Report and Literature Review

**DOI:** 10.3390/jcm11206084

**Published:** 2022-10-15

**Authors:** Mitchell Hunter-Dickson, Patrick Wu, Akshay Athavale, Amanda Ying Wang

**Affiliations:** 1Department of Renal Medicine, Concord Repatriation General Hospital, Sydney, NSW 2139, Australia; 2Department of Renal Medicine, Royal North Shore Hospital, St Leonards, NSW 2065, Australia; 3Concord Clinical School, Faculty of Medicine and Health, The University of Sydney, Sydney, NSW 2006, Australia; 4Renal and Metabolic Division, The George Institute for Global Health, The University of New South Wales, Sydney, NSW 2052, Australia; 5The Faculty of Medicine and Health Sciences, Macquarie University, Sydney, NSW 2109, Australia

**Keywords:** angiomyolipoma, everolimus, genetics, cystic kidney disease, tuberous sclerosis

## Abstract

We report a case of misdiagnosed tuberous sclerosis complex (TSC) in a patient without TSC gene variant presenting with bilateral renal angiomyolipomas and seizures in the context of strong family history of polycystic kidney disease. Clinical diagnosis of tuberous sclerosis complex was made and treatment with everolimus reduced size of renal angiomyolipomas. In this case, report we discuss the association between tuberous sclerosis complex and polycystic kidney disease and novel treatment for TSC.

## 1. Introduction

Tuberous sclerosis complex (TSC) is an autosomal dominant tumour suppressor gene disorder characterised by the development of hamartomatous and cystic growths in multiple organs including the central nervous system (CNS), skin, heart, kidneys and lungs [1]. TSC has an estimated incidence of 1 per 6000–10,000 live births, however, due to variable expression this may be an underestimate [2]. Typically, the disorder presents with seizures, and in its more severe forms, with severe cognitive impairment. Manifestations of TSC in the kidneys involve development of multiple enlarging angiomyolipomas (AMLs), simple kidney cysts, polycystic kidneys and atypical renal cell carcinoma [3]. AMLs are benign tumours composed of bloods vessels, smooth muscle and adipose tissue and can occur in 70–80% of patients with TSC [2,3]. We report an atypical presentation of TSC in a young male without cognitive impairment found to have CNS and kidney involvement necessitating treatment with everolimus.

## 2. Case Report

A 21-year-old male university student of Chinese heritage presented to an Australian teaching hospital with intermittent bilateral flank pain, more pronounced on the right in 2017. Medical history was significant for febrile convulsions and focal seizures during adolescent years. There was a reported history of polycystic kidney disease (PKD) and mild hypertension. He was not taking any prescribed or over the counter medications. Family history was significant for polycystic kidney disease in his father and paternal grandfather, the latter of which required renal replacement therapy in his 7th decade. 

Clinical examination revealed hypertension with blood pressure of 140/85 mmHg. The remainder of the cardiorespiratory examination was unremarkable. Abdominal examination revealed non-tender palpable kidneys bilaterally, without clinical evidence of polycystic changes. There were two hypopigmented lesions on the anterior chest wall measuring 15 mm × 0.5 mm that did not fluoresce under Wood’s Lamp examination. There was one hyperpigmented lesion on the lower trunk that was not well circumscribed. Bedside ophthalmological examination was undertaken, identifying likely retinal hamartomas, however, their presence was unable to be confirmed through formal ophthalmological assessment.

Initial biochemical evaluation demonstrated normal kidney function with creatinine of 80 µmol/L and estimated glomerular filtration rate (eGFR) > 90 mL/min/1.73 m^2^. Urine studies demonstrated no proteinuria, hematuria or leukocyturia and was negative for bacterial growth. Kidney ultrasound identified normal sized kidneys at 10.9 cm (left) and 10.4 cm (right). Multiple, bilateral AMLs were identified with the largest measuring 8 cm in diameter in the right kidney. Multiphase contrast enhanced computerised tomography (CT) of the abdomen confirmed ultrasound findings and the dominant AML underwent angioembolisation. Following the procedure, flank pain resolved completely. Progress abdominal CT three months later showed a significant reduction in the size of the largest AML, measuring 3 cm in diameter (Figure 1).

Due to loss to follow-up, the patient was next reviewed by his nephrologist after three years. In the interim, he had developed focal seizures affecting his right hand and was referred to a neurologist. Neurological evaluation included an electroencephalogram (EEG) that demonstrated moderate amplitude, symmetrical and visually responsive 9–10 Hz alpha rhythm and low amplitude irregular 18–25 Hz beta activity fronto-centrally. There was increased alpha activity with 3 min of hyperventilation and no change on photic stimulation, there was no epileptiform activity or asymmetry. Magnetic resonance imaging (MRI) of the brain showed three areas of focal cortical thickening in keeping with cortical tubers the largest measured 25 mm × 18 mm (Figure 2). One cortical tuber was located near the motor strip on the left, accounting for right sided focal seizures. Due to significant multi-system involvement including cortical tubers, kidney AML and hypo- and hyper- pigmented skin lesions, the criteria for diagnosis of TSC were met and the patient was referred for genetic testing. Given the family history of PKD and clinical features of TSC, initial genetic studies utilising massive parallel sequencing assessed for contiguous deletion at *16p 13.3*, but were negative. Subsequent testing using Multiplex ligation-dependent probe amplification for causative genes *TSC1* and *TSC2* were also negative, potentially secondary to somatic mosaicism for pathogenic TSC gene variants, intronic variants or non-hereditary causes.

The patient was again lost to follow due to the impacts of the coronavirus (COVID-19) pandemic. He re-presented to the outpatient nephrology clinic with right sided flank pain in April 2022, with no associated hematuria. A kidney ultrasound demonstrated enlargement of one of the previously known AMLs, now measuring 5.8 cm in diameter. He commenced treatment with everolimus 5 mg daily, a mammalian target of rapamycin (mTOR) inhibitor with evidence of decrease in AML size. On repeat imaging 6 weeks after treatment initiation, the lesion had reduced to 4.8 cm (Figure 3). Kidney function remained normal throughout. He tolerated everolimus well with good compliance to treatment, experiencing only minor adverse effects including mild acne and oral ulcerations, which resolved spontaneously. After an initial trough everolimus concentration of 5.1 ng/mL, the dose was increased to 10 mg daily with no significant increase in adverse effects. Three monthly kidney ultrasound was performed to evaluate effectiveness of treatment and AML size. 

## 3. Discussion and Literature Review

We have presented a mild case of TSC mosaicism with an atypical clinical history and negative genetic testing. TSC is a heterogenous disorder with a spectrum of clinical manifestations. Most commonly, it presents with seizures, often with severe cognitive impairment and variable involvement of other organs [1]. While AML is a rare cause of end stage kidney disease, spontaneous AML bleeding is the major cause of mortality in TSC [4]. TSC is autosomal dominant with the causative genes identified as *TSC1* and *TSC2*, however, up to two-thirds of patients have de novo mutations and 15–20% of patients have no genetic variant identified through standard genetic testing, as was seen in our case [5]. Failure to identify pathogenic genetic variants on standard testing may represent genetic mosaicism at the *TSC1*/*TSC2* loci or intronic variants. Next generation sequencing and skin tumor DNA assessment of patients with negative initial genetic testing found genetic variants in 85%, including mosaicism in 58% and intronic mutations in 40% [5]. *TSC1* and *TSC2* encode Hamartin and Tuberin respectively. These proteins form a heterodimer that integrates cell signals to regulate mTOR activity [1]. Unregulated mTOR activation leads to increased cellular growth, cell division and angiogenesis, leading to tumour development [3].

The diagnostic criteria for TSC were developed in 2012 and updated in 2021 [6]. The criteria consist of a set of major and minor manifestations [6]. Major criteria include cerebral lesions, kidney, cardiac and pulmonary lesions, as well as dermatological manifestations. Minor criteria are primarily dermatological features but also include simple kidney cysts and non-renal hamartomas [1]. In the updated diagnostic criteria, diagnosis can also be made through identification of pathogenic variants in *TSC1* or *TSC2* even without clinical features, though identification of variants is not necessary to make the diagnosis [6], as in our patient.

Kidney manifestations are common in patients with TSC, including development of enlarging AMLs. AMLs exceeding 3 cm in diameter are at significantly increased risk of spontaneous rupture, leading to retroperitoneal haemorrhage in up to 21% of patients [1]. TSC is infrequently associated with development of chronic kidney disease, however, if multiple surgical procedures are required to manage AMLs, there can be loss of kidney parenchyma and associated impaired kidney function. Severely impaired kidney function may occur if there is coexisting TSC and autosomal dominant polycystic kidney disease (ADPKD) as can occur in 2–3% of patients with TSC2. The *TSC2* gene is found in close association to the ADPKD gene *PKD1*. Patients with contiguous deletion at *16p13.3* typically present with more severe features of TSC and PKD [3]. Under these circumstances, impaired kidney function often develops early in the third decade of life [3]. One of the challenges in diagnosing TSC in this case arose due to the patient’s previous diagnosis of PKD. There are a number of differential diagnoses for macroscopic cystic renal lesions including ADPKD, TSC, Hippel-Lindau syndrome and *DICER1* mutation [7]. In TSC, the initial assessment may be complicated by substantial phenotypic variability in renal manifestations, with cysts of varying size and number and AMLs with variable fat content [1,7]. Particularly in the contiguous deletion syndrome, patients may have only cysts, without identifiable AMLs [7].

The mainstay of TSC management has traditionally been supportive, involving monitoring for enlargement of subependymal giant cell astrocytomas (SEGAs), a common neurological manifestation, and kidney AMLs to thresholds necessitating surgical intervention. However, three recent studies EXIST-1, 2 and 3 and their long-term outcome data have confirmed the role of mTOR inhibitors in reducing tumour growth rates in SEGAs, AMLs and dermatological manifestations [4,8]. Further, mTOR inhibitors have been demonstrated to improve epilepsy control in patients with treatment resistant seizures [9]. These randomised controlled trials were published between 2013 and 2016, recruiting a total of 601 patients with TSC (Table 1). 

EXIST-1 confirmed the role of everolimus in treating SEGAs in children. This study included 117 patients randomised 2:1 to everolimus titrated to a trough concentration of 5–15 ng/mL versus placebo. Thirty-five percent of the everolimus group reached the primary endpoint of a 50% reduction in volume of SEGA compared to 0% in the placebo arm. Reductions in tumour volume were evident by 12 weeks on MRI and no cases of progression of SEGA were seen in the everolimus group [8]. The open-label extension phase of this trial demonstrated persistent response over 192 weeks with 62% reaching primary endpoint [10].

EXIST-2 included 118 patients with mean age 32 randomised 2:1 to everolimus at a standard dose of 10 mg/day versus placebo. Twenty-nine percent of the patients included had a kidney AML exceeding 8 cm in diameter while almost 40% had undergone previous intervention. By weeks 24 and 96, 55% and 64.5% of treated patients had reached primary efficacy endpoint of at least a 50% reduction from baseline in sum of volumes of target AML lesions, compared to 0% of the placebo group [4,11]. Eighty percent of patients receiving everolimus had at least a 30% reduction in AML size compared to 3% of the placebo group. Extension trials of EXIST-2 showed eGFR to remain stable in patients treated with everolimus with reduced rates of renal adverse events. No patients receiving everolimus experienced AML related bleeding during the extension study [11]. Similar results were seen in a Chinese population of 18 patients with TSC. After 12 months of treatment with everolimus, 66.7% of patients had achieved a greater than 50% reduction in the sum of volume of the largest AML lesions [12]. 

EXIST-3 explored the role of adjunctive everolimus in treatment-resistant seizures associated with TSC. This study included 366 patients (median age 10) with treatment-resistant epilepsy randomised 1:1:1 to placebo, low-exposure everolimus (target trough 3–7 mg/mL) or high-exposure everolimus (target trough level 9–15 ng/mL). This study showed a dose–response relationship with a 25% or greater reduction in seizure frequency observed in 37.8% of patients in the placebo arm compared to 52.1% in the low-exposure and 70% in the high-exposure arm. Regression modelling in this study demonstrated that every 15 days of treatment resulted in a 4.8% reduction in seizure frequency. This effect may be related to mTOR inhibitors exhibiting antiepileptic effects by altering signaling pathways and protein expression [9].

There is significant side-effect burden from mTOR inhibitors. Common adverse effects include stomatitis, nasopharyngitis, headache, cough, hypercholesterolemia and acne-like skin lesions [13]. The majority of the adverse events in EXIST trials were grade 1–2 with the most common being stomatitis, occurring in up to 65% of patients, typically within the first month of therapy. Other common adverse events included hypercholesterolemia and infection, though similar infection rates were reported in treatment and placebo arms [4,8]. In EXIST-2, urinary tract infections were the most common infective adverse event with rates of 15%, while in EXIST-3, upper respiratory tract infections were most common with rates of 13–15% reported [4,9]. The majority of adverse events were managed with everolimus dose reduction and occurred at lower frequency after 12 months [11]. Including extension trials, 8–13% of patients ceased everolimus due to adverse events [6,10,14]. Everolimus and other mTOR inhibitors have the potential for drug interactions. Everolimus is a substrate of cytochrome P450 3A4, 3A5 and 2C8 and is also a substrate for the efflux pump P-glycoprotein [15]. A number of antiepileptic agents including carbamazepine and phenytoin are potent inducers of CYP3A416 which may result in failure to achieve adequate everolimus drug exposure and therapeutic failure [16]. Close monitoring of trough concentrations will be required in this circumstance. 

## 4. Conclusions

We have presented a complex case of TSC with negative genetic studies, diagnosed clinically with kidney AML and CNS manifestations. Current evidence suggests that treatment of kidney AML with oral mTOR inhibitor everolimus be commenced when AML size exceeds 3 cm in diameter. Imaging of these lesions should be performed annually and angioembolisation is recommended as second line treatment [6]. There is now strong evidence for everolimus in treatment of other manifestations of TSC including SEGAs and epilepsy, however, everolimus trough concentration monitoring may be necessary due to the propensity for drug–drug interactions. It is important that clinicians be aware of TSC as differential diagnosis in patients with presenting with cystic kidney disease and be familiar with its treatment and potential complications. 

## Figures and Tables

**Figure 1 jcm-11-06084-f001:**
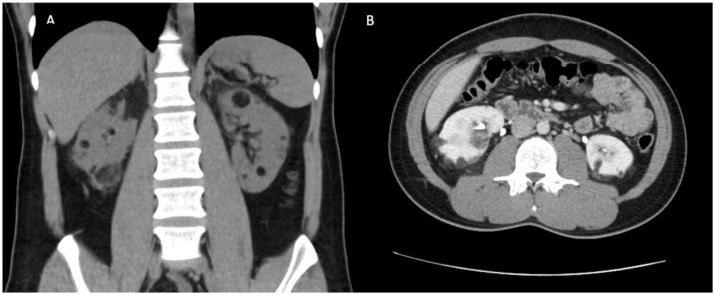
Coronal (**A**) and axial (**B**) contrast computed tomography post angioembolisation of largest AML, demonstrating presence of multiple kidney angiomyolipomas, largest measuring 3 cm in diameter.

**Figure 2 jcm-11-06084-f002:**
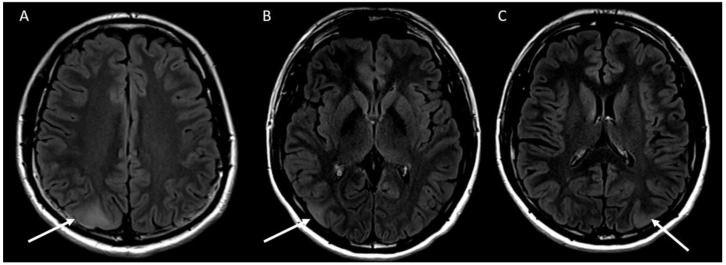
(**A**) T2/FLAIR hyperintensity with associated focal cortical thickening in the right parietal lobe measuring 25 × 18 mm. (**B**) T2/FLAIR hyperintensity at the right parieto-occipital junction measuring 19 × 10 mm. (**C**) T2/FLAIR hyperintensity at the left parieto-occipital junction measuring 14 × 10 mm.

**Figure 3 jcm-11-06084-f003:**
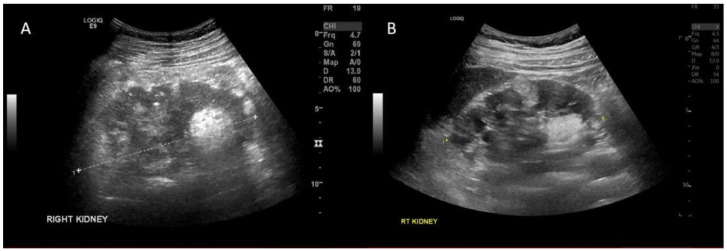
Kidney ultrasound demonstrating reduced size in right kidney lower pole angiomyolipoma before and after everolimus treatment. (**A**)—Pre-treatment [5.8 cm in diameter] (**B**)—Post-treatment [4.8 cm in diameter].

**Table 1 jcm-11-06084-t001:** Summary of EXIST randomised control trials of everolimus in TSC.

Study	Patients	Age (Everolimus Group)	Randomisation	Inclusion Criteria	Primary Outcome	Adverse Events
EXIST-1 [8] (SEGA)	117	1–24 Median 9.5	2:1 everolimus versus placebo, target trough concentration 5–15 ng/mL	At least one SEGA with diameter ≥1 cm on MRI that was enlarging, or new lesion or hydrocephalus	>50% reduction in total volume of all SEGAs Reached by 42% at 24 weeks, 3% in placebo	Mouth ulceration 32% Stomatitis 31% Convulsions 23% Pyrexia 22%
EXIST-2 [4] (AML)	118	18–61 Median 32	2:1 everolimus 10 mg daily versus placebo	Age >18 with at least one AML 3 cm or larger in longest diameter Definite diagnosis of TSC or sporadic LAM Excluded if had had complication or intervention on AML in prior 6 months	≥ 50% reduction in AML volume (sum of volume of all target AMLs identified at baseline) By week 24 Reached by 55% for everolimus, 0% for placebo	Stomatitis 48% Nasopharyngitis 24% Acne-like skin lesions 22% Headache 22% Cough 20% Hypercholesterolaemia 20%
EXIST-3 [9] (Seizures)	366	2–56 Median 10.1	1:1:1 to: Low exposure (LE) target: 3–7 ng/mL High exposure (HE) target: 9–15 ng/mL Placebo	TSC and treatment resistant epilepsy 16 or more seizures in 8 week baseline phase, receiving 1–3 antiepileptic drugs	25% reduction in seizure frequency 37% in placebo arm 52.1% in LE group (median 29.3% reduction seizure frequency) 70% in HE group (median 39.6% reduction)	LE: Stomatitis 55% (0% grade 3 or 4) Diarrhoea 17% HE: Stomatitis 64% (4% grade 3 or 4) Diarrhoea 22%

Abbreviations: AML, angiomyolipoma; LE, low exposure, HE, high exposure; TSC, tuberous sclerosis complex; LAM, Lymphangioleiomyomatosis; SEGA, subependymal giant cell astrocytoma; MRI, magnetic resonance imaging.

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
