# Peer review of "Management of Renal Angiomyolipomas in Tuberous Sclerosis Complex: A Case Report and Literature Review"

_jcm, 2022, doi:10.3390/jcm11206084_

Round 1

Reviewer 1 Report

This is a great case which gives perspective to the overlapping presentation of TSC and PKD. Overall authors do a good job in describing the case and discussing the relevant literature to the topic.

Major points:

- It would be important to provide more details regarding the diagnostic work up. This includes providing representative images of the abdominal CT demonstrating the size of the renal AMLs. 

- Provide comments on EEG in terms of how the background looked and if there were any other electrographic changes on the EEG. Provide duration of EEG, if sleep was captured, was hyperventilation and/ or photic stimulation performed. Was there any asymmetry in the background of the EEG as would be expected if there are significant tubers around one hemisphere vs other.

- Provide location of the tubers, number of tubers, and the areas that they may be affecting, and whether it localized to the patient's clinical symptoms. Providing a representative MRI image of the brain showing cortical tubers is also important.

- Authors should comment on the responsiveness to the higher dose of Everolimus, patient compliance to the medication, if data is available.

- Authors conclude that the absence of TSC pathogenic variant means that patient had mosaicism. However several other possibilities have previously been studied. The authors do not have evidence of mosaicism here, and hence other hypotheses like cell-specific responses to the variant, developmental, environmental factors, and modifier gene effects need to be discussed. Some of these can be found in the Tuberous Sclerosis Complex Working Group papers like Sahin et al 2016 Pediatric Neurology.

- The term "mutation" has largely been replaced with "variant" with subclassification of "pathogenic or non-pathogenic variant", when referring to changes seen upon human DNA sequencing, For appropriate nomenclature please refer to Richards et al 2015 which is the recommendation from ACMG.

Reviewer 2 Report

The authors present a well written manuscript.  The topic of the study is very important, as the authors aim to describe clinical presentation and diagnostic difficulties of adult patient newly diagnosed with tuberous sclerosis and finally treated with mTOR inhibitor and review of the literature.

However I have few comments and questions:

1.      What kind of genetic testing was performed to diagnosed TSC1/TSC2 genes and confirmed NMI status in this case?

2.      You have stated in the first sentence of Discussion that You “have presented a mild case of TSC mosaicism with an atypical clinical history and negative genetic testing”. In my opinion in this case it was only suspicion of mosaicism, it was not confirmed by genetic testing. According to Tyburczy et al. (PLOS Genetics 2015) for detection of mosaicism NGS analysis with high read depth is required and analysis of TSC-related tumors to detect mutations in TSC1/TSC2 in TSC individuals.

3.      There is a mistake in line 114-115: “coexisting TSC and autosomal dominant polycystic kidney 114 disease (ADPKD) as can occur in 2-3% of patients with TSC3”. There is no TSC3.

4.      Did You monitor Everolimus trough concentration in this case?

5.      The “Conclusions” in my opinion should be rewritten. The first two sentences are not conclusions but only summary what you have done.

Reviewer 3 Report

(case report summary) 

This case reports an adjacent gene syndrome of TSC and polycystic kidney in an autosomal dominant family. The figure shows that oral everolimus treatment for concomitant AML was effective for AML after a 6-week follow-up.

(Informed and consent)

First, there is no indication that informed consent was obtained from the patient for the submission of this case to JCM. Even if it is a single report, this case is an autosomal dominant disease with a family history, and information on the father and grandfather must be included, and the case must have been submitted with the patient's consent.

(case presentation) 

Unfortunately, this report is not informative as a case report. Only one case is covered, and the paper is rather devoted to previous literature reports in the second half of the discussion. As this is a case report, not a comprehensive review, a more careful reporting of this case should be done.

As indicated by the family history, this case has polycystic kidneys in the background. It is assumed that this is a case history of concomitant AML. The extent of polycystic kidneys as well as AML is an issue in the prognosis of the kidneys, but there are no MRI or other imaging findings of polycystic kidneys.
   The message the authors wanted to emphasise in the management of AML in the title was not conveyed.
   The MRI should be presented and the size and findings of the nodules classified and described. There are also many reports showing efficacy in post-treatment changes in nodule size and epilepsy, and brain MRI/CT should be followed up.
   There is mention of partial epilepsy and normal EEG, but no mention of the frequency or type of epilepsy.

Ophthalmological findings are not described. What were the ophthalmological findings?

The skin rash should be described with proper dermatological terminology and size.

In addition, everolimus treatment is now becoming the standard of care worldwide, but no trough blood values are given. What is the rationale for increasing the everolimus dose without measuring trough values when AML is shrinking?

Genetic testing, which is of interest, is not properly pursued until the final diagnosis. If a single case of this neighbouring gene is reported, I would like to see a little more pursuit through whole-genome analysis and comprehensive genetic analysis.

Overall, a more detailed case report and new findings would be considered a report with novelty, but this report does not reach that level. Discussion and case summaries are easy to understand. How about a case report with a few cases or a more detailed review article?

Round 2

Reviewer 3 Report

Previously, the reviewers made a slightly stronger academic criticism of this case report. This time, however, the authors made appropriate revisions to the article and resubmitted it promptly. The reviewers appreciate this academic attitude. We also appreciate the ethical importance of the revised description of informed consent.

Now, if there is no ophthalmology consultation, please add a statement of fact: "We have not confirmed the hamartoma".

It is unfortunate that no MRI images are presented. However, I understand about the correction of the text. If the images are available and maintainable, please reconsider.

Finally, there is mention of TSC3 in the discussion. Can the author(s) add an adequate definition or explanation of the concept of TSC3 more in detail to this article? If so, the reviewers believe that TSC3 should be mentioned in both the introduction and the discussion. The authors are encouraged to consider adding this TSC3 addition.

Best regards,

Dr. Reviewer
